# Inserting CTL Epitopes of the Viral Nucleoprotein to Improve Immunogenicity and Protective Efficacy of Recombinant Protein against Influenza A Virus

**DOI:** 10.3390/biology13100801

**Published:** 2024-10-07

**Authors:** Marina Shuklina, Liudmila Stepanova, Olga Ozhereleva, Anna Kovaleva, Inna Vidyaeva, Alexandr Korotkov, Liudmila Tsybalova

**Affiliations:** Smorodintsev Research Institute of Influenza, Ministry of Health of the Russian Federation, 15/17 Prof. Popova Str., St. Petersburg 197376, Russia

**Keywords:** influenza, universal vaccine, M2 ectodomain, HA2, NP, flagellin, recombinant protein

## Abstract

**Simple Summary:**

The influenza virus is a global problem for humanity due to its high variability. Research is ongoing worldwide to develop a so-called universal vaccine to control this infection. Chimeric proteins are one of the platforms for the development of such a vaccine. Within this platform, the least variable proteins of the influenza virus are used so that the vaccine can protect against many strains. In this work, we investigated the possibility of enhancing the efficacy of a candidate vaccine protein by introducing an additional fragment of influenza virus into the construct. Antigen-specific antibody and T-cell immune responses in mice to vaccination were evaluated. Animals were challenged with a lethal dose of influenza viruses to assess the efficacy of the candidate vaccine proteins. It was found that the introduction of new conserved antigens into the construct affects the formation of antibodies to other vaccine antigens but does not affect the protective efficacy of the candidate protein.

**Abstract:**

Conserved influenza virus proteins, such as the hemagglutinin stem domain (HA2), nucleoprotein (NP), and matrix protein (M), are the main targets in the development of universal influenza vaccines. Previously, we constructed a recombinant vaccine protein Flg-HA2-2-4M2ehs containing the extracellular domain of the M2 protein (M2e) and the aa76–130 sequence of the second HA subunit as target antigens. It demonstrated immunogenicity and broad protection against influenza A viruses after intranasal and parenteral administration. This study shows that CD8+ epitopes of NP, inserted into a flagellin-fused protein carrying M2e and HA2, affect the post-vaccination immune humoral response to virus antigens without reducing protection. No differences were found between the two proteins in their ability to stimulate the formation of follicular Th in the spleen, which may contribute to a long-lasting antigen-specific humoral response. The data obtained on Balb/c mice suggest that the insertion of CTL NP epitopes into the flagellin-fused protein carrying M2e and HA2 reduces the antibody response to M2e and A/H3N2. In C57Bl6 mice, this stimulates the formation of NP-specific CD8+ Tem and virus-specific mono- and multifunctional CD4+ and CD8+ Tem in the spleen and completely protects mice from influenza virus subtypes A/H1N1pdm09 and A/H3N2.

## 1. Introduction

Recombinant proteins are one of the most common platforms for creating broad-spectrum vaccines against influenza viruses [1]. These vaccines offer undeniable advantages: They are safe to produce; are well tolerated by vaccinated people; have very few medical contraindications; and can reduce hospitalization, complications, and deaths [2,3,4,5]. The recombinant proteins used in vaccines currently undergoing clinical trials contain both B-cell and T-cell epitopes of viral proteins.

Numerous studies have shown that the presence of antigen-specific T cells correlates with a decrease in viral load, disease severity, and mortality during influenza infections in both humans and model animals [6,7]. Moreover, heterosubtypic immunity and breadth of protection may be associated with T cells targeting conserved influenza antigens [8,9].

The internal viral proteins, such as nucleoprotein (NP) and matrix protein M1, contain many T-cell epitopes and have a much lower mutation rate than hemagglutinin (HA) or neuraminidase (NA), the primary targets of the antibody response [10,11]. Conserved influenza virus proteins, including the hemagglutinin stem domain (HA2), NP, and M, are currently the main targets in the development of cross-protective (universal) influenza vaccines [12]. 

NP is a type-specific and highly conserved protein that remains over 90% identical among different viral strains [12]. NP contains immunodominant CD8+ T-cell epitopes and induces cross-protection against influenza A viruses [13]. NP is the main antigen recognized by cytotoxic T lymphocytes (CTLs) during influenza infection. NP peptides are presented by MHC class I molecules on the surface of virus-infected cells, thereby helping to destroy infected cells and preventing the spread of viral infection.

A number of predicted and experimentally validated CD8+ T-cell epitopes have been identified within the NP protein, and their association with human and mouse histocompatibility antigens has been shown [6,14,15,16,17]. 

NP fragments are used in several universal influenza vaccines currently undergoing different stages of clinical trials (phase III: Multimeric-001 (M-001); phase II: MVA-NP+M1, OVX836, FLU-v; phase I: VGX-3400, INO-3401, FP-01.1), according to the Universal Influenza Vaccine Technology Landscape database [1]. An unregistered vaccine M-001 includes two CTL NP epitopes: aa 335–350 and aa 380–393. The FLU-v peptide vaccine contains the consensus NP sequence of influenza A viruses (aa 255–275) [18]. These vaccines have demonstrated their safety and efficacy, as well as their ability to induce an NP-specific T-cell response.

Previously, we designed a recombinant vaccine protein Flg-HA2-2-4M2ehs using the extracellular domain of the M2 protein (M2e) and the aa76–130 sequence of the second HA subunit (HA2) as target antigens. Flagellin (Flg) is a natural ligand of Toll-like receptor (TLR) 5 and exhibits a strong adjuvant property when administered together with foreign antigens by different routes of administration [19]. The ability of Flg to serve both as a platform and an adjuvant for vaccine development has been demonstrated for various model infections including influenza [20,21,22,23]. The Flg-HA2-2-4M2ehs protein was shown to be immunogenic following intranasal and parenteral administration. It induced the production of specific local and systemic antibodies, the antigen-specific CD4+ and CD8+ response, and the protection against various influenza A viruses (H1, H2, H3, H5, and H7) [24,25].

In this study, we introduced two CTL epitopes of NP into the recombinant protein Flg-HA2-2-4M2ehs. We investigated the protective properties of the two proteins (with and without NP epitopes) and compared humoral responses in a mouse model by estimating the formation of effector memory CD4+ and CD8+ T cells (Tem) and follicular CD4+ cells (Tfh) in secondary lymphoid organs.

## 2. Materials and Methods

### 2.1. Recombinant Proteins

The two recombinant proteins used in this study, namely Flg-HA2-2-4M2ehs and Flg-HA2-2-NP335-NP255-4M2ehs, were designed and obtained in collaboration with the Department of the Molecular Biology of Microorganisms at the Federal Research Center “Fundamentals of Biotechnology”, Russian Academy of Sciences.

The fusion protein Flg-HA2-2-4M2ehs contained full-length flagellin from *S. typhimurium* with a histidine tag at the N-terminus, the consensus HA2 sequence (aa 76–130) of influenza A virus phylogenetic group 2 (A/H3N2, A/H7N9), and four tandem copies of M2e (M2eh-M2es-M2eh-M2es) at the C-terminus [24,26]. In Flg-HA2-2-NP335-NP255-4M2ehs, fragments of the NP protein (NP255 and NP335) were inserted between the consensus HA2-2 sequence and four tandem copies of M2e (Figure 1A). The proteins were produced in *E. coli* strain DLT1270 using the pQE30 Flg_HA2_NP335-NP255_4M2ehshs 2his and pQE30 Flg_HA2_4M2ehshs 2his vectors. Both proteins were purified using metal affinity chromatography on a Ni sorbent under denaturing conditions. The final dialysis solution was PBS.

The following peptides were included in the fusion proteins: 

-M2eh (SLLTEVETPIRNEWGSRSNDSSD)—the consensus sequence of the extracellular domain of the M2 protein of the human influenza A viruses;-M2es (SLLTEVETPTRSEWESRSSDSSD)—the M2e sequence of influenza A/California/07/09 (H1N1pdm09) virus;-HA2-2 (RIQDLEKYVEDTKIDLWSYNAELLVALENQHTIDLTDSEMNKLFE KTRRQLRENA)—a consensus sequence of the conserved region of HA2 (aa 76–130) of influenza A/H3N2 and A/H7N9 viruses;-NP255 (aa 255–275) DLIFLARSALILRGSVAHKS—a highly conserved region with more than 70% identity; a consensus sequence for influenza A viruses. It contains many human CTL epitopes, confirmed both experimentally (HLA-A 02, 03, 11) and theoretically (HLA-A 24, HLA-B 07, 08, 27), as well as T-cell epitopes of mice [27]; -NP335 (aa 335–350) SAAFEDLRVLSFIRGY—a highly conserved region, homologous among influenza A viruses of subtypes H1, H2, H3, and H9. It contains human CTL epitopes, confirmed both experimentally (HLA-A 02, 03, HLA-B 18, 27, 37, 44) and theoretically (HLA-A 01, 26, HLA-B 15, 40, 58), as well as T-cell epitopes of mice [28].

The consensus sequence of the M2e peptide of the human influenza A virus was modified by replacing two cysteines with serines to prevent protein aggregation due to the formation of disulfide bridges. As has been shown previously, this modification does not lead to a change in the immunogenicity of M2e [29].

Protein 3D modeling was performed using the Phyre2 server [30] and visualized using USCF Chimera [31].

### 2.2. Sodium Dodecyl Sulfate–Polyacrylamide Gel (SDS-PAGE) Electrophoresis and Western Blot

SDS-PAGE electrophoresis was carried out under reducing conditions according to the standard method [32]. Electrophoresis was performed at 12 mA until the dye front (bromphenol blue) reached the lower edge of the gel. The gel was stained in Coomassie G-250 solution overnight under rolling. After staining, the gel was washed in bidistilled water. The gel was documented using the ChemiDoc MP System (Bio-Rad, Hercules, CA, USA). For Western blot analysis, bands were electrotransferred to a nitrocellulose membrane (Bio-Rad, USA). Membranes were blocked with 3% BSA overnight at room temperature, and protein bands were detected by membrane staining with rabbit polyclonal antibodies specific to bacterial flagellin (Abcam, Cambridge, UK) or mouse anti-M2e monoclonal antibody 14C2 (Abcam, Cambridge, UK). Staining with anti-NP and anti-HA antibodies was not performed; incubations with antibodies were performed for 1 h in PBS containing 0.1% Tween-20 and 3% BSA, and then the samples were washed. Bands were visualized by staining the membrane for 1 h at room temperature with peroxidase-labeled secondary antibodies, goat anti-rabbit IgG (Invitrogen, Waltham, MA, USA), or goat anti-mouse IgG (Abcam, UK). Finally, blots were incubated in TMB Immunoblot solution (Invitrogen, USA) for 5 min (Figure 1C).

### 2.3. Mice

Female BALB/c (H-2d) and C57Bl/6 (H-2b) mice (16–18 g) were purchased from the Stolbovaya mouse farm at the State Scientific Center of Biomedical Technologies, Russian Academy of Medical Sciences. The animals were housed at the vivarium of the Smorodintsev Research Institute of Influenza in accordance with current regulations. The experiments were approved by the Bioethics Committee on the Use of Animals of the Smorodintsev Research Institute of Influenza under protocol No. 58/08/22. 

### 2.4. Immunization

Female BALB/c mice (16–18 g) were immunized subcutaneously with 10 μg/100 μL of the recombinant protein Flg-HA2-2-4M2ehs or Flg-HA2-2-NP335-NP255-4M2ehs, three times with two-week intervals. The control group of mice was injected with 100 μL of PBS. Serum samples were collected from five mice per group after the second boost to determine antibody titers (Figure 1). To study the formation of follicular helper T cells (Tfh), spleens and lymph nodes (LNs) were collected from five mice per group on the 7th day after the second boost.

To study T-cell response, female C57Bl6 mice (H-2b) (16–18 g) were immunized in the same way. Two weeks after the second boost, the animals were euthanized in a CO_2_ chamber (Vet Tech Solutions, Tamil Nadu, India), and spleens were isolated (Figure 2).

### 2.5. Isolation of Cells from the Spleen and Lymph Nodes

Spleens and regional lymph nodes (pooled superficial cervical and brachial lymph nodes) from the control and immunized mice were harvested after the third immunization. The animals were euthanized using a Vet Tech Solutions CO_2_ box.

Mouse splenocytes and lymph nodes were mechanically ground and purified from cell debris by filtration through a syringe filter with a 70 μm pore size (Syringe Filcons, BD Biosciences, Franklin Lakes, NJ, USA). Erythrocytes were lysed with RBC lysing buffer (Biolegend, San Diego, CA, USA). Cells were washed with RPMI-1640 complete medium containing 10% HI FBS, 100 U/mL penicillin, and 100 μg/mL streptomycin. Cell concentrations were adjusted to 1 × 10^6^ cells/mL.

### 2.6. Serum Antibody Detection by ELISA 

ELISA (enzyme-linked immunosorbent assay) was performed as previously described [24]. Briefly, 96-well microtiter plates (Greiner Bio-One GmbH, Kremsmünster, Austria) were coated with synthetic M2e peptides (5 μg/mL) or influenza viruses A/Aichi/2/68 (H3N2) and A/California/07/09 (H1N1pdm09) (1 μg/mL). Polyclonal HRPO-labeled goat anti-mouse IgG, IgG1, IgG2a, and IgA (Abcam, UK) were used. TMB (BD Bioscience, USA) was used as a substrate. The optical density (OD) was measured using a Multiskan SkyHigh microplate reader (Thermo Fisher Scientific, Waltham, MA, USA) at a wavelength of 450 nm. The titer was determined as the highest serum dilution with an OD at least twice that of the blank’s average value. 

### 2.7. Flow Cytometry and Intracellular Cytokine Staining (ICS) 

The ICS assay was performed as described earlier [15]. Splenocytes were harvested at day 7 or 14 after the second boost and were stimulated for 6 h at 37 °C with 10 μg of the M2eh (peptide) and NP peptides and for 24 h with 1 μg of influenza virus in the presence of 1 μg/mL of Brefeldin A (BD, USA) and purified hamster anti-mouse CD28. Staining with the surface marker CD107a was carried out simultaneously with the activation. The cells were incubated with Zombie Aqua and subsequently stained with CD3e-FITC, CD8a-APC-Cy7, CD4-PerCP, CD62L-PE-Cy7, and CD44-APC antibodies (Biolegend, USA) and then were permeabilized using the BD Cytofix/Cytoperm Plus protocol (BD Bioscience, Franklin Lakes, NJ, USA) and stained with anti-TNF-α-BV421, anti-IFN-γ-PE, and IL-2-BV711 (Biolegend, USA). The gating strategy is shown in Appendix A.

Lymph nodes were harvested on day 7 post-second boost. Briefly, 1 × 10^6^ cells were stained with Zombie Aqua (Zombie Aqua™ Fixable Viability Kit, Biolegend, USA), to enable the gating of live cells during the analysis, and with surface markers CD4-PerCP-Cy5.5, CXCR-5-APC, CD8-APC/Cy7, and CD44-BV510 (Biolegend, USA) for 30 min at 4 °C. Then, cells were washed with DPBS + 2% HI FBS and resuspended in DPBS. A description of the gating tactics with an indication of the phenotype of the studied populations is shown in Appendix A. 

### 2.8. Challenge with Influenza Viruses

For lethal challenge experiments, mouse-adapted influenza strains A/Aichi/2/68 (H3N2) and A/California/07/09 (H1N1pdm09) were used in a dose of 10 LD50. Fourteen days after the third administration (Figure 1), mice (*n* = 10/group) were challenged intranasally (i.n.) under inhalation anesthesia (2–3% isoflurane mixed with 30% oxygen (O_2_) and 70% nitrous oxide (N_2_O)). The animals were monitored for survival and weight loss daily for two weeks.

### 2.9. Statistical Analysis 

Flow cytometry data were analyzed using Kaluza 2.2 software (Beckman Coulter, Brea, CA, USA). Statistical analysis was performed using GraphPad Prism, version 10.2.2. To compare the groups, one-way ANOVA was used. Significant differences in survival among mouse groups were analyzed using the Mantel–Cox test. Antibody and T-cell response data were graphically presented using Tukey plots. 

## 3. Results

### 3.1. Characterisation of Recombinant Proteins

Protein 3D modeling was performed using the Phyre2 server. The tertiary structure of the flagellin protein is shown as an example. The native secondary structure and the alpha helix of the HA2-2 fragment are better preserved in the Flg-HA2-2-4M2ehs construct in contrast to Flg-HA2-2-NP335-NP255-4M2ehs (Figure 1B). These structural aspects may affect the antibody response to recombinant proteins. 

Both proteins demonstrated electrophoretic mobility in SDS-PAGE: ~100 kDa. Protein integrity was confirmed in Western blot using anti-M2e and anti-Flg antibodies (Figure 1C).

### 3.2. Antigen-Specific Antibody Response in Sera of BALB/c Mice after Immunization

To evaluate the antibody response to the chimeric proteins, conserved peptides M2eh and M2es and influenza viruses (A/California/07/09 H1N1pdm09 and A/Aichi/2/68 H3N2) were coated onto high binding plates (1 µg/mL), and IgG or IgA titers were measured by ELISA (Figure 3A–F). After the second boost with recombinant proteins, M2e-specific antibody titers significantly increased compared to the control group (Figure 3A,B). However, the levels of IgG and IgA to M2eh were lower in mice immunized with Flg-HA2-2-NP335-NP255-4M2ehs compared to Flg-HA2-2-4M2ehs (Figure 3A,D). Moreover, we observed a shift in the IgG2a/IgG1 ratio toward IgG2a in mice immunized with Flg-HA2-2-NP335-NP255-4M2ehs (Figure 3C). 

Flg-HA2-2-4M2ehs elicited significantly higher virus-specific serum IgG titers (anti-HA2-2 IgG) than Flg-HA2-2-NP335-NP255-4M2ehs (Figure 3E,F). It is hypothesized that the inclusion of NP epitopes affects the tertiary structure of the recombinant protein, and this may affect the ability of the antibody to bind to the whole virus.

### 3.3. Population of Tfh Cells in Spleens and Lymph Nodes in BALB/c Mice

We compared the ability of two recombinant proteins to stimulate the formation of Tfh cells (CD4+CD44+CXCR5+) in lymph nodes and the spleen on day 7 after the third immunization. There were no changes in the percentages of lymph node Tfh cells compared to control mice (Figure 4A). However, a significant increase in the number of Tfh was detected in the spleens of mice on day 7 after the second boost with either protein (Figure 4B).

### 3.4. Antigen-Specific T-Cell Response in Spleens in C57Bl6 Mice

To evaluate antigen-specific T-cell response after subcutaneous immunization with the recombinant proteins Flg-HA2-2-4M2ehs and Flg-HA2-2-NP335-NP255-4M2ehs, splenocytes from five C57Bl6 mice per group were obtained two weeks after the second boost. The splenocytes were re-stimulated with the M2e peptide (M2eh), A/H3N2 influenza virus, and a pool of NP peptides (NP335 + NP255). The percentages of M2e-, H3N2-, and NP-specific CD4+ and CD8+ effector memory T cells (Tem and CD44+/CD62L-), intracellular cytokine production (IL-2, TNFα, and IFN-γ), and CD107a+ expression were evaluated.

Immunization with both proteins did not change the level of M2e-specific CD8 Tem, but the administration of Flg-HA2-2-4M2ehs resulted in the appearance of a significant number of M2e-specific CD4 Tem (Figure 5A,B). The recombinant protein Flg-HA2-2-4M2ehs stimulated the emergence of M2e-specific CD8+ Tem producing TNF-α (Figure 5A) and CD4+ Tem producing either IL-2 or both IL-2 and IFN-γ (Figure 5B). The absence of M2e-specific CD4+ or CD8+ Tem, producing intracellular cytokines, was revealed in the case of the Flg-HA2-2-NP335-NP255-4M2ehs protein. NP-specific IFN-γ producing CD8+ Tem was observed in the spleens of mice immunized with the Flg-HA2-2-NP335-NP255-4M2ehs protein (Figure 5C).

C57Bl6 mice immunized with Flg-HA2-2-4M2ehs or Flg-HA2-2-NP335-NP255-4M2ehs showed a reliable increase in the population of virus-specific CD4+Tem and CD8+Tem compared to the control group (Figure 6A,B). These Tem cells produced only IFN-γ. No significant differences were found in the number of virus-specific Tem between the two proteins.

The re-stimulation of mouse spleen cells by influenza A/H3N2 virus led to a significant increase in the number of CD8+ effector memory T cells expressing the CD107a+ marker of degranulation after immunization with both proteins (Figure 6C). The simultaneous expression of CD107a+, coupled with intracellular IFN-γ synthesis, was observed in CD4+ and CD8+ Tem in the case of Flg-HA2-2-4M2ehs and only in CD8+ Tem after immunization with Flg-HA2-2-NP335-NP255-4M2ehs (Figure 6D).

### 3.5. Protection against Lethal Influenza Challenge

Two weeks after the second boost, Balb/c mice were challenged intranasally with mouse-adapted viruses A/Aichi/2/68 (H3N2) and A/California/07/09 (H1N1pdm09). Body weight and survival were monitored for 14 days. Immunization with both of the recombinant proteins provided complete protection (mouse survival rates were 90–100%) (Figure 7). In groups immunized with recombinant proteins, maximum weight loss was 12–15%, and animals started to recover on days 7–8 post-challenge with either virus. Mice immunized with Flg-HA2-2-NP335-NP255-4M2ehs experienced more weight loss compared to Flg-HA2-2-4M2ehs in A/California/07/09 infection (Figure 7A). Control animals demonstrated more pronounced body weight loss (20–22%) after the challenge in both cases.

## 4. Discussion

The development of multi-antigen universal influenza vaccines is a way to achieve broad protection. Influenza surface proteins are widely used as targets in cross-protection vaccine development, including the stem region of HA2 and the extracellular domain of M2 (M2e). However, surface proteins are more susceptible to escape mutations [33]. In contrast, nucleoprotein (NP) is an internal structural protein. It is conserved among multiple influenza subtypes and demonstrates a low mutation rate throughout the virus evolution. Using this protein as a target antigen has become one of the main strategies for the development of cross-protective influenza virus vaccines [34,35]. 

These vaccine antigens elicit different types of immune responses. Non-neutralizing antibodies induced by HA2 can promote antibody-dependent cellular cytotoxicity (ADCC) and CD4+ and CD8+ T-cell response related to cross-protection against influenza viruses [25,32,36,37,38,39]. While anti-M2e antibodies do not prevent influenza virus infection, they can eliminate infected cells through complement-dependent cytolysis, antibody-dependent cellular cytotoxicity, and/or antibody-dependent cellular phagocytosis [40,41,42]. M2e-specific CD4+ T-cell response is also formed [38,42]. Different M2e vaccine constructs have demonstrated cross-protection [43,44,45]. The main mechanism of the immune response to NP is the activation of the CD8+ cytotoxic T lymphocytes through the presentation of T-cell epitopes in complex with MHC-I. CD8+ T lymphocytes eliminate virus-infected cells, inhibit virion release, control virus replication, reduce the severity of flu-related symptoms, and mediate cross-protection against heterologous influenza viruses [35,46,47,48,49,50,51]. 

Multi-antigen universal influenza vaccines can employ various platforms. NP/M2e nanoparticles induce robust NP-specific T-cell and M2e-specific serum antibody responses and increase survival rates in BALB/c mice during viral challenges [50]. Vaccination with the recombinant protein 3M2e–3HA2–NP strongly induces influenza antigen-specific antibody response, stimulates cytotoxic T lymphocytes, and confers complete protection against homologous and heterologous influenza viruses in mice [49]. Wang et al. [52] constructed protein nanoparticles incorporating two conserved influenza antigens, NP and NA. Immunization with these nanoparticles induced significant antigen-specific humoral (neutralizing antibodies) and cellular responses (IFN-γ and IL-4-secreting cells and NP147–155 tetramer-specific CTL responses) and conferred heterosubtypic influenza protection. The recombinant virus-based influenza vaccines carrying full-length NP fused with four copies of M2e (RVJ-4M2eNP) provided NP- and M2e-specific humoral and cellular immune responses in mice, as well as the highest cross-protection in mice challenged with 20 MLD50 of PR8 [53]. The authors believe that cross-protection correlates with both NP- and M2e-specific humoral and cellular immune responses induced by RVJ-4M2eNP. 

The present study has shown that the recombinant protein Flg-HA2-2-4M2ehs can induce a pronounced antibody response to the vaccine antigens HA2 and M2e. However, when two T-cell epitopes of NP (NP255–275 and NP335–350) were inserted into the recombinant protein Flg-HA2-2-4M2ehs, the level of M2e- and virus-specific IgG reduced, and the IgG2a/IgG1 ratio shifted toward IgG2a.

It is known that follicular Th cells (Tfh) are essential for the formation of a strong humoral response, ensuring protection against a wide range of pathogens [54,55,56]. They localize in secondary lymphoid organ follicles and mediate B-cell immunoglobulin affinity maturation, class switch recombination, and the generation of long-lived plasma cells and memory B cells [57,58,59,60]. Influenza infection induces strong, long-term protective antibody response and the formation of antigen-specific Tfh, whereas most influenza vaccines do not [61]. During influenza infection, virus antigens are presented to CD4+ T cells for over 7 days [62,63]. After immunization with an inactivated vaccine, the presentation of viral antigens is observed within 3 days [64]. The short period of antigen presentation and the absence of a pronounced inflammatory response may explain why Tfh formation and antibody response to inactivated vaccines are significantly lower than those induced by natural infection [65]. However, repeated immunizations may enhance the development of antigen-specific Tfh and prolong humoral immunity [66,67].

Comparative analysis of memory T cells in the spleens of mice after subcutaneous immunization with Flg-HA2-2-4M2ehs or Flg-HA2-2-NP335-NP255-4M2ehs showed that the insertion of NP peptides into the protein Flg-HA2-2-4M2ehs results in a weaker T-cell response to M2e and the formation of the NP-peptide-specific CD8+IFN-γ+ cells. In this case, the virus-specific response is the same for both proteins and is characterized by the formation of CD4+ and CD8+ Tem monoproducers of IFN-γ. However, while after immunization with the Flg-HA2–2-4M2ehs protein, only CD8+CD107a+ and CD8+CD107a+IFN-γ+ were detected, immunization with Flg-HA2-2-NP335-NP255-4M2ehs induced both CD8+ and CD4+ Tem cells, demonstrating cytotoxic potential (degranulation marker CD107a+). The expression of CD107a+ coupled with IFN-γ synthesis was also observed, i.e., these T cells are multifunctional, which correlates with milder disease and protection against lethal influenza infection [68,69]. 

Thus, the insertion of two CD8+ epitopes of the NP protein into Flg-HA2–2-4M2ehs significantly impacts the immune response to M2e and HA2-2 antigens attached to the carrier protein flagellin. This can be attributed to a change in the tertiary structure of the protein, which may affect the strength of the antigen-specific humoral response. In addition, a change in the IgG2a/IgG1 ratio was observed for Flg-HA2-2-NP335-NP255-4M2ehs with an increase in the proportion of IgG2a antibodies that are more effective in antibody-dependent cellular cytotoxicity than IgG1. Immunization with Flg-HA2-2-NP335-NP255-4M2ehs was characterized by the formation of NP-specific CD8+ Tem producers of IFN-γ and virus-specific CD4+ and CD8+ IFN-γ+ and CD107a double producers with cytotoxic potential. Despite the differences in the immune response induced by the studied proteins, experiments on the protective effectiveness showed that both recombinant proteins almost completely protected mice (90–100% survival) from virus subtypes A/H1N1pdm09 and A/H3N2 in a dose of 10 LD50. In future studies, we plan to utilize transgenic mice to evaluate the potential efficacy of these CD8 epitopes in humans.

## 5. Conclusions

This study demonstrates that CD8+ epitopes of the NP protein inserted into a flagellin-fused protein affect the post-vaccination humoral immune response to virus antigens but do not reduce protection. The two proteins do not differ in their ability to promote the formation of follicular T cells, which can contribute to a long-lasting antigen-specific humoral response. It can be concluded that CTL epitopes of the NP protein inserted into the flagellin-fused protein containing M2e and HA2 lead to a reduced antibody response to M2e and A/H3N2 but stimulate the formation of NP-specific CD8+ Tem and virus-specific mono- and multifunctional CD4+ and CD8+ Tem in spleens, completely protecting mice from influenza virus subtypes A/H1N1pdm09 and A/H3N2.

## Figures and Tables

**Figure 1 biology-13-00801-f001:**
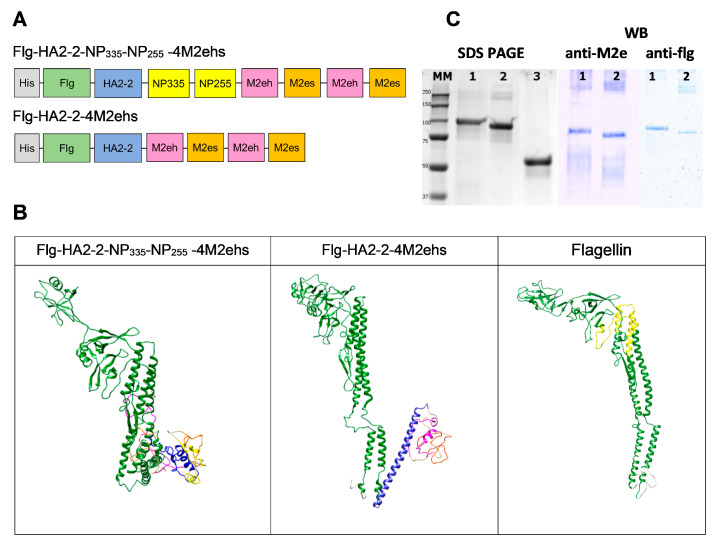
Schematic structure (**A**) and structural 3D models (**B**) of monomeric recombinant fusion proteins Flg-HA2-2-4M2ehs and Flg-HA2-2-NP335-NP255-4M2ehs: orange—M2es, pink—M2eh, blue—the HA2-2 fragment, yellow—the NP fragments, and green—flagellin. SDS-PAGE Coomassie brilliant blue staining and Western blot (**C**) of the recombinant fusion proteins Flg-HA2-2-NP335-NP255-4M2ehs (lane 1), Flg-HA2-2-4M2ehs (lane 2), and Flagellin (lane 3) are also shown.

**Figure 2 biology-13-00801-f002:**
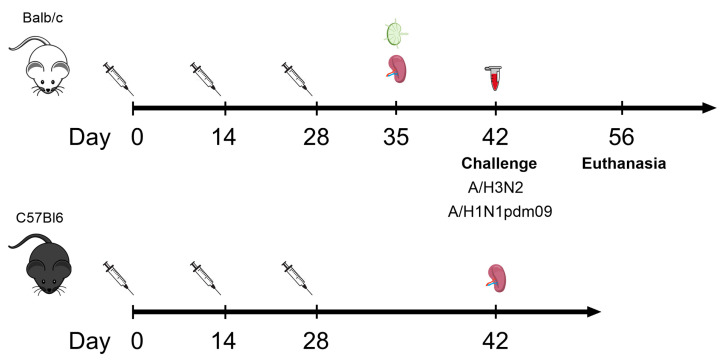
Design of the experiment.

**Figure 3 biology-13-00801-f003:**
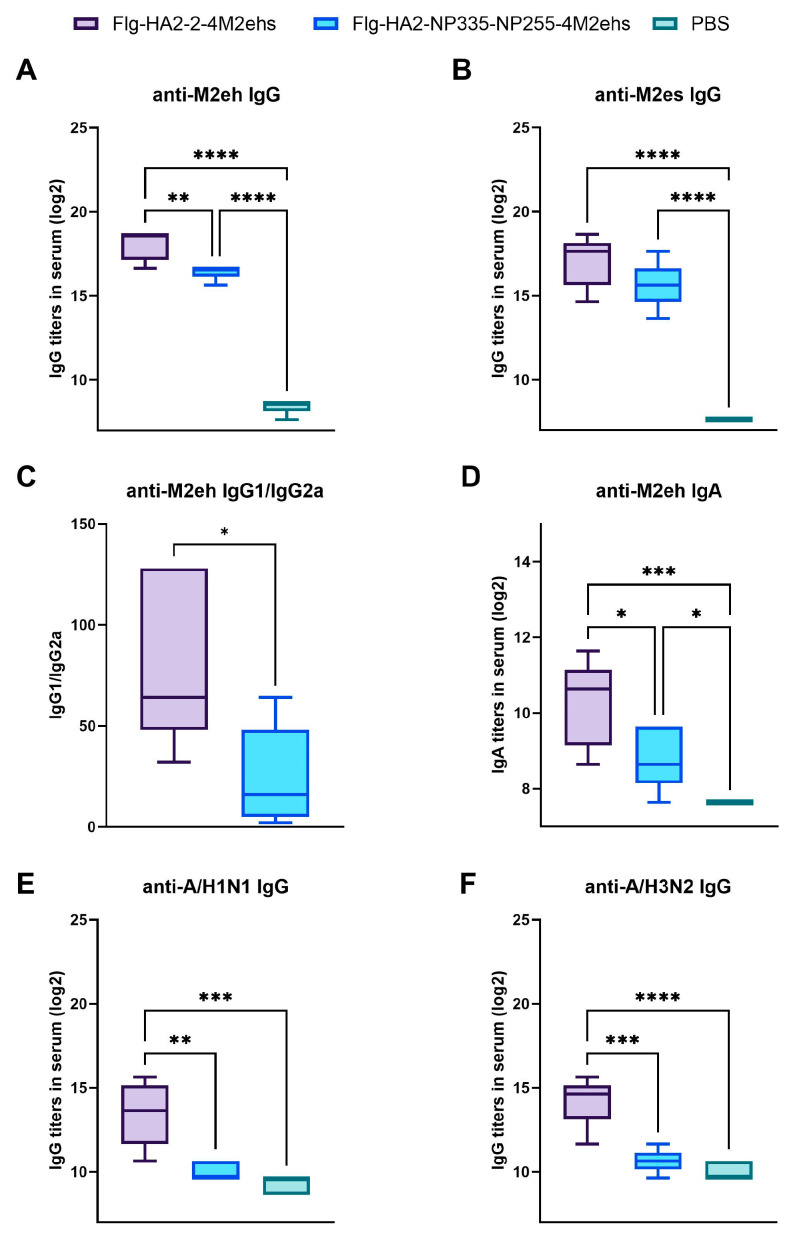
Antibody response in serum (log2 titers). BALB/c mice (*n* = 5/group) were immunized s.c. with 10 μg of Flg-HA2-2-4M2ehs or Flg-HA2-2-NP335-NP255-4M2ehs on days 0, 14, and 28. The control group of mice received PBS. Two weeks after the second boost, M2e-specific IgG (**A**,**B**), anti-M2e IgG subclasses (**C**), IgA (**D**), and virus-specific IgG (**E**,**F**) titers were evaluated by ELISA. The data are presented as Tukey plots. Statistically significant differences between groups are indicated as *—*p* < 0.05; **—*p* < 0.01; ***—*p* < 0.001; and ****—*p* < 0.0001.

**Figure 4 biology-13-00801-f004:**
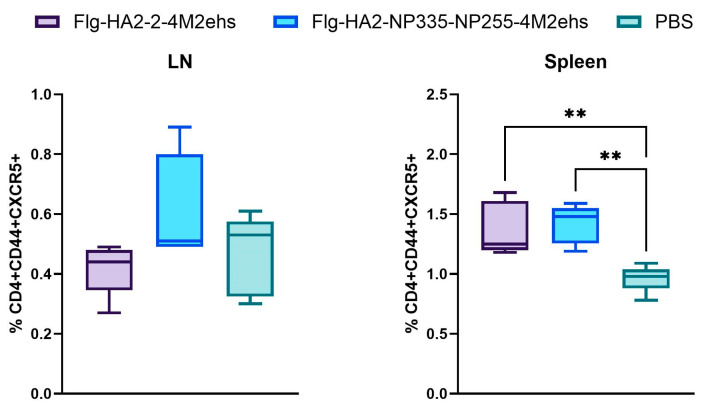
The populations of CD4+CD44+CXCR5+ Tfh cells in lymph nodes (**A**) and spleens (**B**) on day 7 after the third immunization with Flg-HA2-2-4M2ehs or Flg-HA2-2-NP335-NP255-4M2ehs. The data are presented as Tukey plots. *p* values indicate statistically significant differences between groups: **—*p* < 0.01.

**Figure 5 biology-13-00801-f005:**
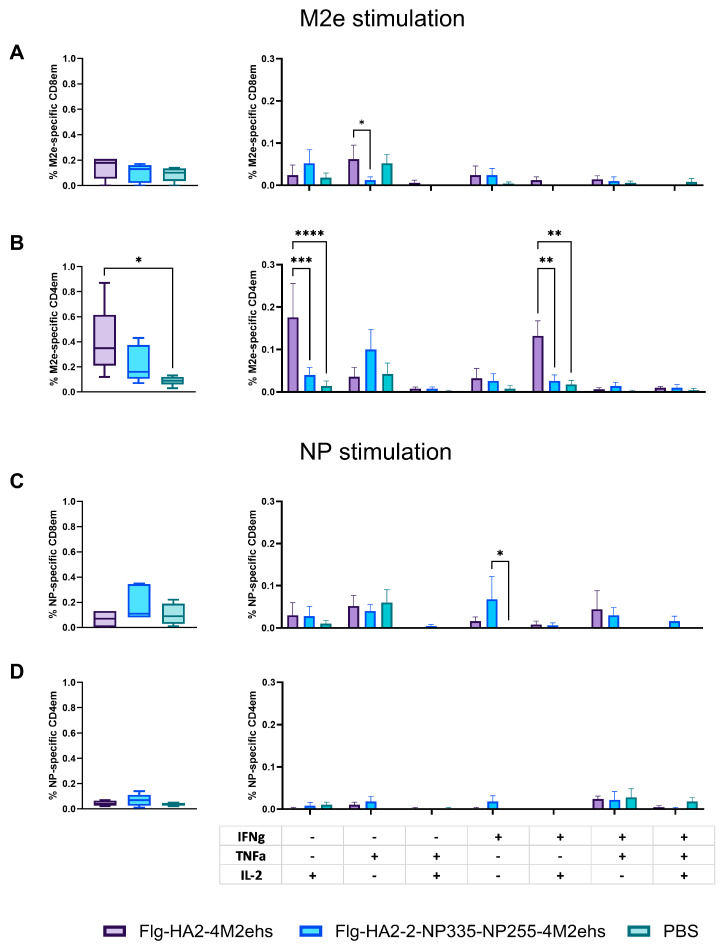
M2e-specific (**A**,**B**) and NP-peptide-specific (**C**,**D**) CD8+ and CD4+ effector memory T cells in spleens and the cytokine profile after subcutaneous immunization of C57Bl6 mice with the recombinant proteins Flg-HA2-2-4M2ehs and Flg-HA2-2-NP335-NP255-4M2ehs. The data are presented as Tukey plots. Significant differences from the control are shown as *—*p* < 0.05; **—*p* < 0.01; ***—*p* < 0.001; and ****—*p* < 0.0001.

**Figure 6 biology-13-00801-f006:**
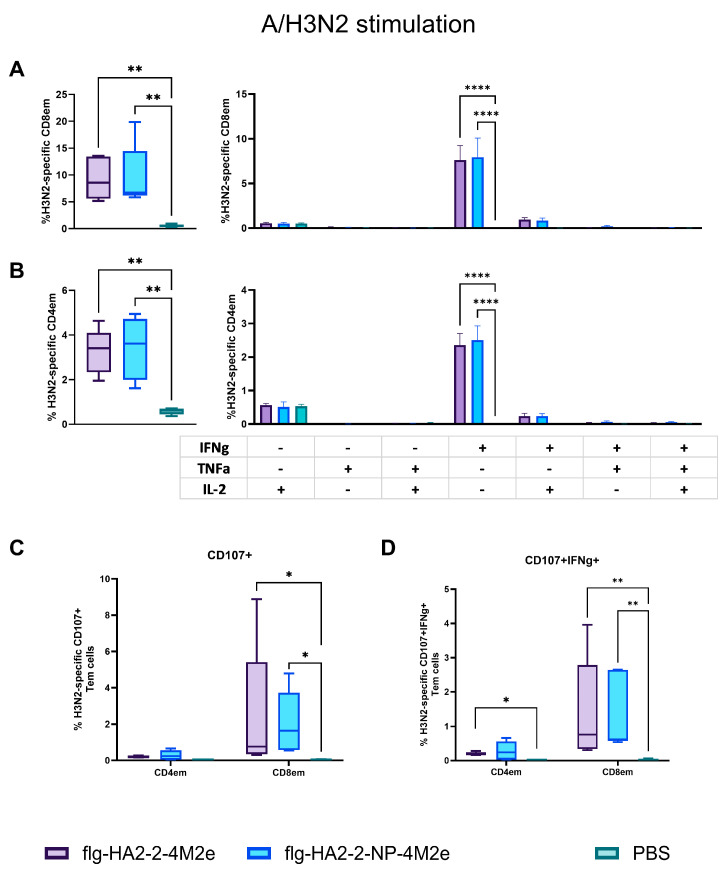
H3N2-specific CD8+ (**A**) and CD4+ (**B**) effector memory T cells and the cytokine profile in spleens after subcutaneous immunization of C57Bl6 mice with the recombinant proteins Flg-HA2-2-4M2ehs and Flg-HA2-2-NP335-NP255-4M2ehs. Expression of the degranulation marker CD107a+ (**C**) and expression of CD107a+ and IFN-γ (**D**) on A/H3N2-specific CD4+ and CD8+ effector memory T cells in the spleen of C57Bl6 mice are also presented. The data are presented as Tukey plots. Significant differences between experimental groups are shown as * —*p* < 0.05; **—*p* < 0.01; and ****—*p* < 0.0001.

**Figure 7 biology-13-00801-f007:**
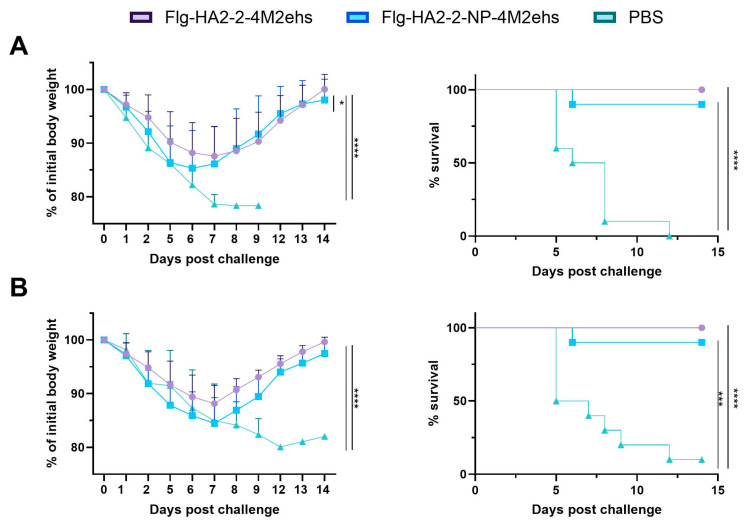
Efficacy of immunization. Balb/c mice (*n* = 10/group) were immunized with fusion proteins Flg-HA2–2-4M2ehs and Flg-HA2-2-NP335-NP255-4M2ehs. Two weeks after the second boost, mice were challenged with 10 LD50 of (**A**) A/California/07/09 (H1N1pdm09) or (**B**) A/Aichi/2/68 (H3N2). Body weight (left) and survival rates (right) were monitored daily for 14 days. The *p* values (Mantel-Cox test) indicating statistically significant differences between immunized and control groups are presented as *—*p* < 0.05; ***—*p* < 0.001; and ****—*p* < 0.0001.

## Data Availability

Data can be requested from the correspondent author.

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
