# Peer review of "Inserting CTL Epitopes of the Viral Nucleoprotein to Improve Immunogenicity and Protective Efficacy of Recombinant Protein against Influenza A Virus"

_biology, 2024, doi:10.3390/biology13100801_

Round 1

Reviewer 1 Report

Comments and Suggestions for Authors

Introduction: Suggest authors to review and revise the introduction section (specifically lines 44-73) for brevity and clarity. For example, 

1. Line 32: Please include suitable reference.

2. Line 47-49: "NP peptides are......, thereby helping to destroy infected cells and spread the viral infection". It seems to be missing a word between ‘and’ and ‘spread’. Please review and revise.

3. Line 49-50: "Unlike influenza infection,...". The relevance of this statement here is unclear. Please provide some context.

4. Line 55-57: Please specify that this statement is for influenza vaccines and include appropriate link to the database in the reference

5.  Line 63-65: The authors have described M2e and HA2 proteins in the previously designed contstruct but haven’t mentioned anything about flagellin/ Flg and its purpose as a TLR5 agonist/ adjuvant. Suggest authors to include this information as the new recombinant protein tested in this paper also include Flg.  

6. Lines 69-73: "In this study, we introduced two CTL epitopes of NP into the recombinant protein Flg-HA2-2-4M2ehs. We investigated the protective properties of the resulting two proteins and compared ..."

This is confusing as it implies that two different proteins were constructed with CTL insertion. One of these two proteins is from the previous study and doesn’t carry CTL epitopes so suggest the authors to revise for clarity.

Methods:

1.    Section 2.1: Information on how the protein was expressed/expression system and purified and buffer system is not included in the methods. The authors have shared this information in the Results; however, such details are more appropriate to be included in methods.

2.    Methodology for SDS PAGE included in Figure 1 is missing.

           3.    Looking at the consensus Me2 of human Influenza A virus, there appears to be a Cys -> Ser modification. Can authors confirm this.

4.    Line 92: Typo - NP335

5.    Can authors clarify why T-cell response was studied in a different mice strain? 

6.    Section 2.7: Suggest authors to add reference to Figure 1 

7. Please reference the supplementary figures in the methods with appropriate supporting information

Results:

1.    Have authors performed any western blot analysis on NP335-NP255 inserted recombinant fusion protein to understand epitope integrity and accessibility?

2.    Is virus-specific IgG response in Fig2 being used as determinant of anti-HA2 response? If yes, please specify in results for clarity.

3.    Figure 5B: Can authors please confirm the significance of difference between the two test groups and the PBS control. The graph shows that the difference between Flg-HA2-NP335-NP255-4M2ehs and PBS is significant but the difference between Flg-HA2-4M2ehs and PBS is not. However, looking at the graph, the readout in Flg-HA2-4M2ehs group appears to even higher the readout in Flg-HA2-NP335-NP255-4M2ehs

4.    Lines 240-241: Figure 5B indicates that there is a significant M2e-specific CD4em response detected in the mice immunized with NP-peptide containing protein

5.     Figure 6B: y- axis label for %H3N2-specific CD4em

6.    Figure 6B and lines 249-255: “The activation of mouse spleen cells by influenza A/H3N2 virus stimulated a significant increase in the expression of the CD107a+ marker on both CD8+ and CD4+ effector memory T-cells after immunization with Flg-HA2-NP335-NP255-4M2ehs, and only on CD8+ Tem after immunization with Flg-HA2-4M2ehs (Fig. 6C). The simultaneous expression of CD107a+, coupled with intracellular IFN-γ synthesis, was also observed in CD4+ and CD8+ Tem after immunization with Flg-HA2-2-NP335-NP255-4M2ehs and only in CD8+ Tem after immunization with Flg-HA2-2-4M2ehs (Fig. 6D).”

Reviewer comment: There seems to be an error in the siginificance markers shown on graphs 6C and 6D, resulting in mismatch between what is highlighted on the graph and the corresponding results discussed in the text. Please review and revise. For example, in Fig 6C, the graph shows that expression of CD107+ marker on CD8/4 in NP protein groups is not significantly different than PBS group but the text above says otherwise. Similar comment for Figure 6D.

Discussion:

1.    Lines 327-328: “The present study has shown that both recombinant proteins (Flg-HA2-2-4M2ehs and Flg-HA2-2-NP335-NP255-4M2ehs) can induce a pronounced antibody response to the vaccine antigens HA2 and M2e. “

Reviewer comment: Authors describe that the present study shows both recombinant proteins induce pronounced antibody response to vaccines antigens HA2 and Me2. However, Fig 3E and 3F, which shows virus-specific IgG response in both protein groups, suggest that the IgG response to HA2 in NP recombinant protein is not significantly different than the PBS control.

2.    Lines 340-342: “During influenza infection, virus antigens are presented to CD4+ T cells for over 7 days [50, 51]. After immunization with an inactivated vaccine, the presentation of viral antigens is observed within 3 days [52]. The short period of antigen presentation and the absence of a pronounced inflammatory response may explain why Tfh formation and antibody response to inactivated vaccines are significantly lower than those induced by natural infection [53]. However, repeated immunizations may enhance the development of antigen-specific Tfh and prolong humoral immunity [54, 55]. We also observed that repeated immunization with both recombinant proteins induces the formation of Tfh, which may contribute to a long-lasting antigen-specific humoral response.”

Reviewer comment:

A.    I am unclear on relevance of lower Tfh formation in inactivated vaccines in this context unless authors have performed a comparison study between an inactivated vaccine form and their recombinant vaccine antigen.

B.    The authors also note that repeated immunizations may enhance the development of antigen-specific Tfh and prolong humoral immunity and that they observed that repeated immunization with their protein induced Tfh formation. In the current study, authors measured formation of Tfh cells only on day 7 after third immunization. Is there data on Tfh formation after 1st or 2nd immunization to understand how repeated immunization impacts Tfh cell level? Without it, it is difficult to conclude if repeated immunization really played a role in levels of Tfh cells observed. 

Author Response

Thank you very much for such a detailed review of our work. You provided a great help in improving our paper.

Introduction: Suggest authors to review and revise the introduction section (specifically lines 44-73) for brevity and clarity. For example, 

  1. Line 32: Please include suitable reference.

Response. Corrected.  “These vaccines offer undeniable advantages: they are safe to produce, are well tolerated by vaccinated people, have very few medical contraindications, can reduce hospitalization, complication and deaths etc. [Pratha Sah , Jorge A. Alfaro-Murillo , Meagan C. Fitzpatrick, Kathleen M. Neuzil , Lauren A. Meyers, Burton H. Singer, and Alison P. Galvani      Future epidemiological and economic impacts of universal influenza vaccines. PNAS, 8, 2019, vol. 116 ,no. 41; Taylor, D.N. Induction of a potent immune response in the elderly using the TLR-5 agonist, flagellin, with a recombinant hemagglutinin influenza-flagellin fusion vaccine (VAX125, STF2.HA1 SI) / D.N. Taylor  [et al.] / Vaccine. – 2011. – Vol. 29. – P. 4897-4902; Turley, C.B. Safety and immunogenicity of a recombinant M2e-flagellin influenza vaccine (STF2.4xM2e) in healthy adults / C.B. Turley [et al.] / Vaccine. -2011. – Vol. 29. – P. 5145-5152].”

  1. Line 47-49: "NP peptides are......, thereby helping to destroy infected cells and spread the viral infection". It seems to be missing a word between ‘and’ and ‘spread’. Please review and revise.

Response. CorrectedNP peptides are presented by MHC class I molecules on the surface of virus-infected cells, thereby helping to destroy infected cells and preventing spread the viral infection.”

  1. Line 49-50: "Unlike influenza infection,...". The relevance of this statement here is unclear. Please provide some context.

Response This statement was removed

  1. Line 55-57: Please specify that this statement is for influenza vaccines and include appropriate link to the database in the reference

Response. Corrected.NP fragments are used in several universal influenza vaccines currently undergoing different stages of clinical trials (phase III: Multimeric-001 (M-001); phase II: MVA-NP+M1, OVX836, FLU-v; phase I: VGX-3400, INO-3401, FP-01.1), according to the Universal Influenza Vaccine Technology Landscape database [https://ivr.cidrap.umn.edu/universal-influenza-vaccine-technology-landscape].”

  1. Line 63-65: The authors have described M2e and HA2 proteins in the previously designed contstruct but haven’t mentioned anything about flagellin/ Flg and its purpose as a TLR5 agonist/ adjuvant. Suggest authors to include this information as the new recombinant protein tested in this paper also include Flg. 

Response. “Flagellin is a natural ligand of Toll-like receptor (TLR) 5 and exhibits a strong adjuvant property when administered together with foreign antigens by different route of administration [Bates JT, Honko AN, Graff AH, Kock N, Mizel SB. Mucosal adjuvant activity of flagellin in aged mice. Mech Ageing Dev. 2008;129:271–81]. The ability of FliC to serve both as a platform and an adjuvant for vaccine development have been demonstrated for various model infections including influenza [Huleatt JW, Nakaar V, Desai P, Huang Y, Hewitt D, Jecobs A, Tang J, McDonald W, Song L, Evans RK, Umlauf S, Tussey L, Powell TJ. Potent immunogenicity and efficacy of a universal influenza vaccine candidate comprising a recombinant fusion protein linking influenza M2e to the TLR5 ligand flagellin. Vaccine. 2008;26(2):201–14., Song L, Zhang Y, Yun NE, Poussard AL, Smith JN, Smith JK, Borisevich V, Linde JJ, Zacks MA, Li H, Kavita U, Reiserova L, Liu X, Dumuren K, Balasubramanian B, Weaver B, Parent J, Umlauf S, Liu G, Huleatt J, Tussey L, Paessler S. Superior efficacy of a recombinant flagellin:H5N1 HA globular head vaccine is determined by the placement of the globular head within flagellin. Vaccine. 2009;27(42):5875–84, Liu G, Tarbet B, Song L, Reiserova L, Weaver B, Chen Y, Li H, Hou F, Liu X, Parent J, Umlauf S, Shaw A, Tussey L. Immunogenicity and efficacy of flagellin-fused vaccine candidates targeting 2009 pandemic H1N1 influenza in mice. PLoS One. 2011;6(6):e20928, Liu, Ge, et al. "Flagellin-HA vaccines protect ferrets and mice against H5N1 highly pathogenic avian influenza virus (HPAIV) infections." Vaccine 30.48 (2012): 6833-6838.]. “

  1. Lines 69-73: "In this study, we introduced two CTL epitopes of NP into the recombinant protein Flg-HA2-2-4M2ehs. We investigated the protective properties of the resultingtwo proteins and compared ..."

This is confusing as it implies that two different proteins were constructed with CTL insertion. One of these two proteins is from the previous study and doesn’t carry CTL epitopes so suggest the authors to revise for clarity.

Response. Corrected. “In this study, we introduced two CTL epitopes of NP into the recombinant protein Flg-HA2-2-4M2ehs. We investigated the protective properties of the two proteins (with and without NP epitopes) and compared humoral responses in a mouse model by estimating the formation of effector memory СD4+ and СD8+ T cells (Tem) and follicular CD4+ cells (Tfh) in secondary lymphoid organs.”

Methods:

  1. Section 2.1: Information on how the protein was expressed/expression system and purified and buffer system is not included in the methods. The authors have shared this information in the Results; however, such details are more appropriate to be included in methods.

Response. Corrected. Part of results section 3.1 moved to materials and methods section 2.1

  1. Methodology for SDS PAGE included in Figure 1 is missing.

Response. Corrected. Methodology for SDS PAGE is included in the Materials and Methods section 2.2

  1. Looking at the consensus Me2 of human Influenza A virus, there appears to be a Cys -> Ser modification. Can authors confirm this.

Response. The consensus sequence of the M2e peptide of the human influenza A viruses was modified by replacing two cysteines with serines to prevent protein aggregation due to the formation of disulfide bridges. As was shown previously, this modification does not lead to a change in the immunogenicity of M2e [De Filette, M.; Min Jou, W.; Birkett, A.; Lyons, K.; Schultz, B.; Tonkyro, A.; Resch, S.; Fiers, W. Universal Influenza A Vaccine: Optimization of M2-Based Constructs. Virology 2005, 337, 149–161, doi:10.1016/j.virol.2005.04.004. ].

  1. Line 92: Typo - NP335

Response. Corrected

  1. Can authors clarify why T-cell response was studied in a different mice strain? 

Response. Studies of the post-vaccination immune response to immunization of mice with the recombinant proteins were conducted on mice of two lines with different haplotypes: BALB/c (H-2d) and C57Bl/6 (H-2b). According to the literature, laboratory mice of the C57Bl/6 and BALB/c lines are typical representatives of animals with a predominance of Th1 and Th2 types of immune response [            Stewart, D. 2002]. Upon antigen stimulation, BALB/c mice, which have the H-2d haplotype of the major histocompatibility complex, develop a predominantly humoral immune response, while C57Bl/6 mice, which have the H-2b haplotype, are genetically "prone" to the formation of a T-cell immune response [Spellberg В., Edwards J. E. 2000,  Liu, T. et.al 2002]. When mice are infected with an intracellular pathogen (Leishmania major), C57Bl/6 mice demonstrate the formation of a protective Th1-type T-cell response and are resistant to infection, whereas BALB/c mice develop a non-protective Th2 response [Reiner, S. L., Locksley, R. M. 1995, Roch, F., Bach, M. A.1990]. Normally, BALB/c and C57Bl/6 mice have differences in a number of structural and functional indices of the immune system. In BALB/c mice, compared to C57Bl/6, the indices of the volume density of the T-zone of the spleen and the level of IL-2, -4, -10 and TNF-a production by spleen cells are significantly higher. Splenocytes of C57Bl/6 mice are characterized by a higher level of cytotoxic activity. It was shown that BALB/c mice had more CD4+ T cells in the spleen (25.14%) and lymph nodes (56.12–58.91%) and fewer CD8+ T cells (11.65% in the spleen and 20.53–22.88% in the LN) than C57Bl/6 mice (CD4+ 19.85% in the spleen and 40.7–40.9% in the LN; CD8+ 13.98% in the spleen and 26.75–31.0% in the LN) [Xin Chen, et.al. 2005]. Therefore, in order to evaluate all aspects of the formation of the post-vaccination response to immunization with the chimeric protein, mice of different genetic lines were used.

  1. Section 2.7: Suggest authors to add reference to Figure 1 

Response. Corrected

  1. Please reference the supplementary figures in the methods with appropriate supporting information

Response. Corrected

Results:

  1. Have authors performed any western blot analysis on NP335-NP255 inserted recombinant fusion protein to understand epitope integrity and accessibility?

Response. We did a western blot with anti-M2e and anti-Flg antibodies. Unfortunately, we didn't have the opportunity to stain with anti-NP and anti-HA antibodies. Western blot analysis of the proteins was inserted into Fig 2.

  1. Is virus-specific IgG response in Fig2 being used as determinant of anti-HA2 response? If yes, please specify in results for clarity.

Response. Naive mice lacking antiviral antibodies were used in the experiment. Immunization of mice with recombinant proteins containing a fragment of the second HA subunit (aa76-130) with B-cell epitopes resulted in the formation of anti-HA2-2(aa76-130) antibodies detected in ELISA upon sorption of whole viruses onto a plate. Sentence has been supplemented. "Flg-HA2-2-4M2ehs elicited significantly higher virus-specific serum IgG titers (anti-HA2-2 IgG) than Flg-HA2-2-NP335-NP255-4M2ehs (Fig. 3 E, F)."

  1. Figure 5B: Can authors please confirm the significance of difference between the two test groups and the PBS control. The graph shows that the difference between Flg-HA2-NP335-NP255-4M2ehs and PBS is significant but the difference between Flg-HA2-4M2ehs and PBS is not. However, looking at the graph, the readout in Flg-HA2-4M2ehs group appears to even higher the readout in Flg-HA2-NP335-NP255-4M2ehs

Response. Mistake in figure 5B was corrected. Adjusted P Value for   PBS vs. flg-HA2-2-4M2e is 0.0372 and for Flg-HA2-NP335-NP255-4M2ehs is 0.4318.

  1. Lines 240-241: Figure 5B indicates that there is a significant M2e-specific CD4em response detected in the mice immunized with NP-peptide containing protein

Response. Inserted sentenceImmunization with both proteins did not change the level of M2e-specific CD8 Tem, but administration of Flg-HA2-2-4M2ehs resulted in the appearance of a significant number of M2e-specific CD4 Tem (Fig 5AB)”

  1. Figure 6B: y- axis label for %H3N2-specific CD4em

Response Corrected

  1. Figure 6B and lines 249-255: “The activation of mouse spleen cells by influenza A/H3N2 virus stimulated a significant increase in the expression of the CD107a+ marker on both CD8+ and CD4+ effector memory T-cells after immunization with Flg-HA2-NP335-NP255-4M2ehs, and only on CD8+ Tem after immunization with Flg-HA2-4M2ehs (Fig. 6C). The simultaneous expression of CD107a+, coupled with intracellular IFN-γ synthesis, was also observed in CD4+ and CD8+ Tem after immunization with Flg-HA2-2-NP335-NP255-4M2ehs and only in CD8+ Tem after immunization with Flg-HA2-2-4M2ehs (Fig. 6D).”

Reviewer comment: There seems to be an error in the siginificance markers shown on graphs 6C and 6D, resulting in mismatch between what is highlighted on the graph and the corresponding results discussed in the text. Please review and revise. For example, in Fig 6C, the graph shows that expression of CD107+ marker on CD8/4 in NP protein groups is not significantly different than PBS group but the text above says otherwise. Similar comment for Figure 6D.

Response. The error has been fixed. “The re-stimulation of mouse spleen cells by influenza A/H3N2 virus led to significant increase in the number of CD8+ effector memory T-cells expressing the CD107a+ marker of degranulation after immunization with both proteins (Fig. 6C). The simultaneous ex-pression of CD107a+, coupled with intracellular IFN-γ synthesis, was observed in CD4+ and CD8+ Tem in case of Flg-HA2-2-4M2ehs and only in CD8+ Tem after immunization with Flg-HA2-2-NP335-NP255-4M2ehs (Fig. 6D).”

Discussion:

  1. Lines 327-328: “The present study has shown that both recombinant proteins (Flg-HA2-2-4M2ehs and Flg-HA2-2-NP335-NP255-4M2ehs) can induce a pronounced antibody response to the vaccine antigens HA2 and M2e. “

Reviewer comment: Authors describe that the present study shows both recombinant proteins induce pronounced antibody response to vaccines antigens HA2 and Me2. However, Fig 3E and 3F, which shows virus-specific IgG response in both protein groups, suggest that the IgG response to HA2 in NP recombinant protein is not significantly different than the PBS control.

 Response. The sentence in the text has been corrected “The present study has shown that the recombinant protein Flg-HA2-2-4M2ehs can induce a pronounced antibody response to the vaccine antigens HA2 and M2e. “

  1. Lines 340-342: “During influenza infection, virus antigens are presented to CD4+ T cells for over 7 days [50, 51]. After immunization with an inactivated vaccine, the presentation of viral antigens is observed within 3 days [52]. The short period of antigen presentation and the absence of a pronounced inflammatory response may explain why Tfh formation and antibody response to inactivated vaccines are significantly lower than those induced by natural infection [53]. However, repeated immunizations may enhance the development of antigen-specific Tfh and prolong humoral immunity [54, 55]. We also observed that repeated immunization with both recombinant proteins induces the formation of Tfh, which may contribute to a long-lasting antigen-specific humoral response.”

Reviewer comment:

  1. I am unclear on relevance of lower Tfh formation in inactivated vaccines in this context unless authors have performed a comparison study between an inactivated vaccine form and their recombinant vaccine antigen.
  2. The authors also note that repeated immunizations may enhance the development of antigen-specific Tfh and prolong humoral immunity and that they observed that repeated immunization with their protein induced Tfh formation. In the current study, authors measured formation of Tfh cells only on day 7 after third immunization. Is there data on Tfh formation after 1st or 2nd immunization to understand how repeated immunization impacts Tfh cell level? Without it, it is difficult to conclude if repeated immunization really played a role in levels of Tfh cells observed.

Response. Thanks for your comments. Inactivated influenza vaccines are administered once and cause a strain-specific antibody response that is less pronounced than in natural influenza infection. According to the literature, this is associated with a short period of antigen presentation, which leads to weak formation of Tfh. In this work, we did not set the task of comparing the level of Tfh formation in inactivated influenza vaccines and candidate vaccines based on recombinant proteins. Vaccines based on recombinant proteins with conservative low immunogenic antigens are administered repeatedly (2-3 immunizations) to form a pronounced immune response. We do not claim that repeated immunizations really play a role in the formation of a high level of Tfh, since we have not conducted such studies after the first and second immunization. We assume, based on the literature, that this is possible.

Reviewer 2 Report

Comments and Suggestions for Authors

"Inserting CTL epitopes of the viral nucleoprotein to improve immunogenicity and protective efficacy of recombinant protein against influenza A virus" by Shuklina et al., shows the immunogenicity and protection against IAV A(H1N1) and A(H3N2) strains in a mouse model. Authors have made a very systematic approach in this study showing the antibody responses and the enhanced production of CD4+ and CD8+ T cell in the spleen on exposure to the recombinant protein M2ehs.

My suggestions and queries are as follows:

1.       Line 198 states coated with H1N1 and H3N2, explain more detail, what was the concentration? something like no of virus/cm3 etc.

2.       Line 206, based on this study, it is premature to say that the 3D structure is responsible for the higher virus specific IgG production, modify it.

3.       Line 211, explain the reason for selecting 10ug, did the authors try different concentrations? Did they do a dose response study?

4.       In the viral challenge study (Fig 7), did authors monitor the virus copy numbers in BAL or lung tissues? Did they notice a 100% viral clearance with both strains of virus?

Comments on the Quality of English Language

Minor grammer issues, otherwise readable, no major issues detected.

Author Response

Thank you very much for your review and helpful comments!

"Inserting CTL epitopes of the viral nucleoprotein to improve immunogenicity and protective efficacy of recombinant protein against influenza A virus" by Shuklina et al., shows the immunogenicity and protection against IAV A(H1N1) and A(H3N2) strains in a mouse model. Authors have made a very systematic approach in this study showing the antibody responses and the enhanced production of CD4+ and CD8+ T cell in the spleen on exposure to the recombinant protein M2ehs.

My suggestions and queries are as follows:

  1. Line 198 states coated with H1N1 and H3N2, explain more detail, what was the concentration? something like no of virus/cm3

Response. Clarified. 1μg/ml

  1. Line 206, based on this study, it is premature to say that the 3D structure is responsible for the higher virus specific IgG production, modify it.

Response. We hypothesize that the inclusion of NP epitopes affects the tertiary structure of the recombinant protein and this may affect the ability of the anti-HA2 antibody to bind to the whole virus

  1. Line 211, explain the reason for selecting 10ug, did the authors try different concentrations? Did they do a dose response study?

Response. In all works related to recombinant proteins based on flagellin (as a carrier protein and adjuvant), doses of no more than 10 μg are used for immunization (Honko AN 2006, McDonald WF 2007, Huleatt JW 2008, Bates JT 2008, Hong SH, 2012, Delaney KN 2010). In our earlier work we used a dose of 50 μg (Liudmila A. Stepanova et.al. Protection against Multiple Influenza A Virus Strains Induced by Candidate Recombinant Vaccine Based on Heterologous M2e Peptides Linked to Flagellin PLoS ONE 10(3): e0119520. doi:10.1371/journal.pone.0119520). In preclinical studies of the protein Flg-HA2-2-4M2ehs we used doses of 5 and 10 μg/mouse (results not published). No significant differences in immunogenicity were found.

  1. In the viral challenge study (Fig 7), did authors monitor the virus copy numbers in BAL or lung tissues? Did they notice a 100% viral clearance with both strains of virus?

Response. We did not monitor the virus copy numbers in BAL or lung tissues

Reviewer 3 Report

Comments and Suggestions for Authors

The manuscript [biology-3205319], entitled “Inserting CTL epitopes of the viral nucleoprotein to improve 2 immunogenicity and protective efficacy of recombinant protein 3 against influenza A virus” by Prof. Marina Shuklina, et al., reports immunogenicity and broad protection induced by recombinant vaccine protein Flg-HA2-2-4M2ehs against influenza A viruses after intranasal and parenteral administration.

This is an interesting and important study.

Some concerns are listed below for consideration in revision.

Some specific minor concerns:

1.      Line 89: Suggested to revise “homology”  to “identity”. Please check the entire text for the same problem.

2.      Please add the full name for some abbreviated antonyms, such as “ELISA”.

Author Response

Thank you so much for your comments about our work!

Line 89: Suggested to revise “homology”  to “identity”. Please check the entire text for the same problem.

Response. Corrected

Please add the full name for some abbreviated antonyms, such as “ELISA”.

Response. Corrected

Round 2

Reviewer 1 Report

Comments and Suggestions for Authors

1. Lines 75-78: In line 75, I would advise authors to revise as "Flagellin (Flg) is a natural ligand...." so that it is clear that Flg in the recombinant protein is Flagellin. Line 77 mentions FliC - I am assuming authors mean Flagellin here, but the abbreviation has not been defined anywhere in the manuscript and I would recommend keeping it consistent as Flg throughout the manuscript. 

2. Methods section 2.2: Authors have abbreviated the gel method as SDS-PAAG. However, in the corresponding figure 2C, the method has been abbreviated as SDS-PAGE. Suggest authors to keep it consistent throughout and as SDS-PAGE since this is a widely and commonly used acronym for this method.

3. Results:

  1. Have authors performed any western blot analysis on NP335-NP255 inserted recombinant fusion protein to understand epitope integrity and accessibility?

Author's response. We did a western blot with anti-M2e and anti-Flg antibodies. Unfortunately, we didn't have the opportunity to stain with anti-NP and anti-HA antibodies. Western blot analysis of the proteins was inserted into Fig 2.

Reviewer response: Please include this as a gap/ future work in the manuscript. 

Author Response

Comments and Suggestions for Authors

  1. Lines 75-78: In line 75, I would advise authors to revise as "Flagellin (Flg) is a natural ligand...." so that it is clear that Flg in the recombinant protein is Flagellin. Line 77 mentions FliC - I am assuming authors mean Flagellin here, but the abbreviation has not been defined anywhere in the manuscript and I would recommend keeping it consistent as Flg throughout the manuscript. 

Author's response.  Corrected

  1. Methods section 2.2: Authors have abbreviated the gel method as SDS-PAAG. However, in the corresponding figure 2C, the method has been abbreviated as SDS-PAGE. Suggest authors to keep it consistent throughout and as SDS-PAGE since this is a widely and commonly used acronym for this method.

Author's response.  Corrected

  1. Results:
  1. Have authors performed any western blot analysis on NP335-NP255 inserted recombinant fusion protein to understand epitope integrity and accessibility?

Author's response. We did a western blot with anti-M2e and anti-Flg antibodies. Unfortunately, we didn't have the opportunity to stain with anti-NP and anti-HA antibodies. Western blot analysis of the proteins was inserted into Fig 2.

Reviewer response: Please include this as a gap/ future work in the manuscript. 

Author's response. This information was included to methods section.